# Fermentation of the Brown Seaweed *Alaria esculenta* by a Lactic Acid Bacteria Consortium Able to Utilize Mannitol and Laminari-Oligosaccharides

Leila Allahgholi [1,*], Madeleine Jönsson [1], Monica Daugbjerg Christensen [2], Andrius Jasilionis [1], Mehrnaz Nouri [3], Shahram Lavasani [3], Javier A. Linares-Pastén [1], Guðmundur Óli Hreggviðsson [2,4] and Eva Nordberg Karlsson [1]

[1] Division of Biotechnology, Department of Chemistry, Lund University, SE-22362 Lund, Sweden
[2] Matís Ohf, IS-113 Reykjavík, Iceland
[3] ImmuneBiotech AB, Medicon Village, SE-22381 Lund, Sweden
[4] Faculty of Life and Environmental Sciences, University of Iceland, IS-102 Reykjavík, Iceland
[*] Correspondence: leila.allahgholi@biotek.lu.se

**Abstract:** The brown seaweed *Alaria esculenta* is the second most cultivated species in Europe, and it is therefore of interest to expand its application by developing food products. In this study, a lactic acid bacteria consortium (LAB consortium) consisting of three *Lactiplantibacillus plantarum* strains (relative abundance ~94%) and a minor amount of a *Levilactobacillus brevis* strain (relative abundance ~6%) was investigated for its ability to ferment carbohydrates available in brown seaweed. The consortium demonstrated the ability to ferment glucose, mannitol, galactose, mannose, and xylose, of which glucose and mannitol were the most favored substrates. No growth was observed on fucose, mannuronic and guluronic acid. The consortium used different pathways for carbohydrate utilization and produced lactic acid as the main metabolite. In glucose fermentation, only lactic acid was produced, but using mannitol as a carbohydrate source resulted in the co-production of lactic acid, ethanol, and succinate. Xylose fermentation resulted in acetate production. The consortium was also able to utilize laminari-oligosaccharides (DP2-4), obtained after enzymatic hydrolysis of laminarin, and produced lactic acid as a metabolite. The consortium could grow directly on *A. esculenta*, resulting in a pH decrease to 3.8 after 7 days of fermentation. Incubation of the same seaweed in corresponding conditions without inoculation resulted in spoilage of the seaweed by endogenous bacteria.

**Keywords:** *Lactobacillus* culture; fermentation; brown seaweed; laminari-oligosaccharides; metabolite profile

## 1. Introduction

Brown seaweeds constitute about 40% of the global seaweed (macroalgae) production, which amounts to approximately 10 million tons annually. In Europe, around 300,000 tons are harvested with *Laminaria hyperborea*, *Laminaria digitata*, and *Ascophyllum nodosum* as the dominating species [1]. In addition, seaweed cultivation is a growing industry; however, it is still at an early stage in Europe, with total production (of mainly *Saccharina latissima* and *Alaria esculenta*) below 2000 tons annually [1,2]. To advance the transition to a sustainable biobased economy, brown seaweed biomass is of interest as it is abundant in coastal areas of the Atlantic Ocean and can be cultivated and produced in bulk offshore. One area of interest is to expand the use and applications of seaweed by developing new European food products.

As a food resource, brown macroalgae contain a number of potential prebiotic and/or bioactive compounds [3]. Approximately 50% of the seaweed dry weight is composed of carbohydrate compounds, including the sugar alcohol mannitol, and the polysaccharides laminarin, alginate, and fucoidan. Mannitol, a storage carbohydrate in brown seaweed,

constitutes 3–20% of the seaweed dry weight [4], with broad applications in the pharmaceutical and food industries. Mannitol is, however, poorly metabolized in the human body and if consumed in excess can cause diarrhea, abdominal pain, and flatulence [5]. The dietary fiber laminarin is another storage carbohydrate of brown seaweed that comprises 1–25% of the dry weight, dependent on the harvesting time [4]. Laminarin is made of a β-(1,3) glucan with occasional β-(1,6) interchain links [6,7], which in its oligomeric or polymeric forms have potential biomedical applications [1]. Alginate, a structural carbohydrate, is available in the seaweed cell wall. Alginate is a linear polymer made of (1,4)-linked β-D-mannuronic acid (M) and α-L-guluronic acid (G) in different sequences with very well-established applications in the pharmaceutical and food industries. Fucoidan, a sulfated polysaccharide, consists of fucose substituted to different degrees with, e.g., galactose, mannose, xylose, and uronic acids [8]. Fucoidan has been reported to have bioactive properties but is not yet in industrial use.

The dietary fibers in brown seaweeds have shown potential benefits to human health in general and to the gastrointestinal tract in particular [9,10]. The gut microbiota consequently plays an important role in human well-being by fermenting indigestible dietary fibers, supplying essential nutrients, synthesizing vitamin K, saving energy, absorbing ions, and improving the immune system [11]. Human indigenous gut microbes and potentially added probiotics need nutrients to maintain optimal growth and activity. For this purpose, seaweeds are of interest because of their significant number of bioactive compounds. Seaweed dietary fibers, including laminarin, alginate, and fucoidan, have prebiotic properties [7,12], meaning they can be utilized by the human gastrointestinal tract and some probiotic bacteria, providing health benefits for the host.

The many reported health benefits of seaweeds and their potential for sustainable cultivation provide the reasons for the recent growing interest in seaweed for food-related applications in Europe. Seaweed biomass has been a food source in Asia for centuries; however, challenges with post-harvest treatment, food safety, and consumer acceptance have obstructed the introduction of seaweed and seaweed products to new markets [13]. One way to overcome these issues is to ferment seaweed and/or its components with lactic acid bacteria (LAB). This not only alters the sensory properties but also improves shelf life and nutritional value by producing lactic acid and short-chain fatty acids (SCFAs), thereby lowering the pH of the product [14,15].

Another way to utilize this valuable and sustainable biomass is by modification of the seaweed's carbohydrates in the production of value-added ingredients that can be used either as individual health-promoting supplements or as food ingredients that do not alter the taste and the texture of food [16,17].

In this study, we evaluated the potential of a small LAB consortium consisting of three strains of *Lactiplantibacillus plantarum* and one *Levilactobacillus brevis* strain in fermenting the brown seaweed *A. esculenta* by growing the consortium on fresh frozen seaweed biomass, which was either or not pretreated. Furthermore, laminari-oligosaccharides were produced from laminarin as value-added products, and their potential prebiotic properties were evaluated by fermentation implementing the LAB consortium.

## 2. Materials and Methods

### 2.1. Materials

The seaweed species *Alaria esculenta* (L.) Grev. biomass was obtained fresh frozen from the Frøya farm (Seaweed Solutions AB, Trondheim, Norway). *Laminaria hyperborea* (Gunnerus) Foslie used for in-house production of mixed laminari-oligosaccharides was harvested from bay of Breiðafjörður (Íslensk Bláskel & Sjávargróður, Stykkishólmur, Iceland). Individual laminari-oligosaccharides were from *L. digitata* and were purchased from Megazyme (Neogen, Lansing, MI, USA). The LAB consortium, isolated from food, originates from the collections held by ImmuneBiotech AB (Lund, Sweden). All materials were purchased from Sigma–Aldrich (Merck, Rahway, NJ, USA).

### 2.2. Genomic Identification of the Consortium Species and Metagenome Analysis of the Seaweed Biomass

The Genomic DNA from the consortium was extracted by using the GeneJET Genomic Purification Kit (Thermo Fisher Scientific, Waltham, MA, USA) and diluted with nuclease-free water before adding it to the PCR mixture. Gene 16S rRNA was amplified in a total volume of 20 μL. The PCR mixture was prepared by mixing 10 μL of IProof HF master mix (2×) (BioRad, Hercules, CA, USA), 0.5 μM of universal primer pair 16S-fD1 and 16S-rP2 (Integrated DNA Technologies, Leuven, Belgium), 3% (*v*/*v*) DMSO, and the extracted genomic DNA. The PCR was conducted as follows: initial denaturation at 95 °C, 3 min; 34 cycles of 95 °C denaturation, 30 s; annealing at 55 °C, 30 s; and extension at 72 °C, 1 min; with a final extension at 72 °C for 10 min using a T100 thermal cycler (Bio-Rad). The obtained PCR product was analyzed using agarose electrophoresis and subsequently purified using the GeneJET PCR Gel Extraction Kit (Thermo Fisher Scientific) (Figure S1). The purified PCR product was sequenced by Eurofins Genomics (Solna, Sweden). The resulting sequences were analyzed by blastn integrated into the NCBI BLAST suite (blast.ncbi.nlm.nih.gov (accessed on 30 August 2021)) against 16S ribosomal RNA sequences (Bacteria and Archaea) database.

For metagenome analysis of the LAB consortium, the LAB culture was grown on the de Man, Rogosa, and Sharpe (MRS) broth agar plate under an anaerobic condition at 37 °C overnight to obtain the sample of cells. Thereafter, one single colony was inoculated in a liquid MRS medium. Two milliliters of the consortium culture was harvested by centrifugation at $1700 \times g$ for 3 min (Centrifuge 5425, Eppendorf, Hamburg, Germany). The pellet was aseptically washed twice with sterile DNA suspension buffer (10 mM Tris-HCl pH 8.0, 0.1 mM EDTA), and the cells were collected by centrifugation at $1700 \times g$ for 3 min (Centrifuge 5425, Eppendorf). The collected cells were suspended in DNA suspension buffer and then were delivered frozen for metagenome sequencing by CosmosID (Germantown, MD, USA).

The metagenomic analysis of the dried seaweed biomass (for control samples without added inoculum) was analyzed using NovaSeq 6000 (Illumina, San Diego, CA, USA) technology for metagenome sequencing by Eurofins Genomics (Ebersberg, Germany).

### 2.3. Cultivation of the LAB Consortium with Different Carbohydrates Available in Brown Seaweed

2.3.1. Fermentation on the Carbohydrate Available in Brown Seaweed

The MRS broth base was prepared by mixing peptone from casein, tryptic digest 10 g/L, meat extract 10 g/L, yeast extract 5 g/L, Tween 80 1 g/L, $K_2HPO_4$ 2 g/L, Na-acetate 5 g/L, $(NH_4)_3$ citrate 2 g/L, $MgSO_4 \times 7\ H_2O$, 0.2 g/L, and $MnSO_4 \times H_2O$ 0.05 g/L. The prepared MRS base was filled into serum bottles, boiled, and purged with nitrogen gas to maintain the anaerobic condition. The bottles were sealed and autoclaved for 20 min at 1 atm (VE-150 autoclave, Systec, Linden, Germany). The carbohydrate sources were prepared and autoclaved separately. Before the cultivation, an amount of the carbohydrate substrate was added to the bottles to reach the desired concentration. Media were 1% inoculated, and the bottles were incubated at 37 °C (Termaks incubator, Kungsbacka, Sweden). Optical density at 600 nm ($OD_{600}$) was measured every 2 h up to 12 h and then after 24 h and 48 h. Cultivations were made in duplicates.

The natural logarithm of the optical density was plotted against time, and two time points were selected within the exponential growth phase. The specific growth rate was then calculated using the following formula:

$$\mu = ((\text{LnOD}_2) - (\text{LnOD}_1))/(t_2 - t_1)$$

where $\mu$ is the specific growth rate, $OD_2$ is the optical density at time $t_2$, $OD_1$ is the optical density at time $t_1$, and $(t_2 - t_1)$ is the time interval between the two measurements.

### 2.3.2. Seaweed Fermentation

Fresh frozen seaweed biomass was minced by a household mincer while kept ice-cold. Subsequently, 20% (wet biomass (g)/ slurry (g)), corresponding to 2% (dried biomass (g)/ slurry (g)) seaweed slurry, was prepared by mixing the ice-cold seaweed with distilled water in a 500 mL glass reactor (BPC Instruments, Lund, Sweden). The seaweed slurry was subjected to fermentation directly or underwent autoclavation at 1 atm for 20 min prior to the fermentation. Three reactors containing seaweed slurry were prepared for direct fermentation, and three reactors with seaweed slurry were autoclaved at 121 °C for 20 min: two reactors for inoculation with LAB consortium and one reactor as a control without adding any inoculum.

The fermentation inoculum was prepared by pre-culturing the LAB consortium from glycerol stock culture in an MRS broth overnight. The bacteria were reinoculated in MRS broth, cultivated, and harvested at $OD_{600}$ of 2.0 by centrifugation at $3000\times g$ for 20 min at 4 °C (Sigma 3–16PK centrifuge, Sigma Laborzentrifugen, Osterode am Harz, Germany). The supernatant was discarded, and the collected cells were washed twice with 0.9% *(w/v)* saline solution to remove the medium completely. An amount of cell culture corresponding to $1 \times 10^9$ CFU was added to the reactor to reach a concentration of around $2 \times 10^6$ CFU/g slurry. The fermentation with seaweed's endogenous microbiota was performed by fermenting seaweed slurry with and without pretreatment (with or without autoclaving) and without added inoculum. All reactors were airtight and incubated at 37 °C in a water bath while connected to a gas endeavor system equipped with pH electrodes (BPC Instrument). The pH and gas production were monitored for 7 days. Duplicate samples were withdrawn from each cultivation for analysis. Samples were taken every 18 h to monitor the growth in the seaweed slurry by counting CFU (Figure 1).

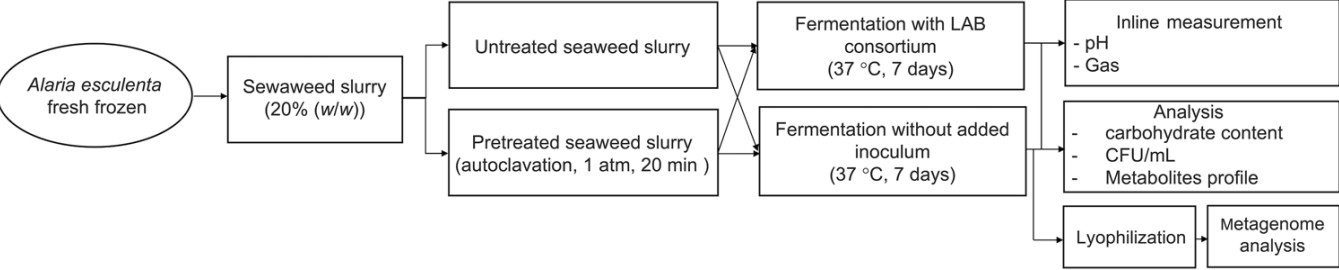

**Figure 1.** Flowchart representing the fermentation experiment. The seaweed slurry (20% *(w/w)*) was subjected to fermentation with LAB consortium and fermentation without any added inoculum.

To analyze carbohydrate consumption and organic acids production, the samples were centrifuged at $3000\times g$ (Centrifuge 5425, Eppendorf) for 10 min at 4 °C. The supernatant was collected, diluted with ultrapure water (Milli-Q grade), and filtered through PTFE membrane 0.2 µm pore size filters (Pall, Port Washington, NY, USA) for further analysis.

### 2.4. Laminari-Oligosaccharide Production

Laminarin content was determined for the brown seaweed species *A. esculenta* and *L. hyperborea*. As the laminarin content of *A. esculenta* was low (2.9% of the dry weight, Table S1), the *L. hyperborea*, with a laminarin content of approximately 20% of the dry weight was used for extraction of laminarin. Laminari-oligosaccharides were subsequently produced from *L. hyperborea* laminarin by enzymatic hydrolysis using a recombinant in-house-produced novel glycoside hydrolase (*Ml*Lam17B) classified under glycoside hydrolase family 17 and originating from *Muricauda lutaonensis* (manuscript in preparation [18]).

### 2.4.1. Laminarin Extraction

Laminarin was extracted from *L. hyperborea* by a two-step water extraction (manuscript in preparation [19]). Seaweed slurry in water (100 g/L) was incubated at 30 °C for 2 h

followed by 2 h incubation at 70 °C with 160 rpm shaking (ES-20/80 shaker incubator, Biosan, Riga, Latvia). The soluble fraction was separated with two-step centrifugation at $5000\times g$ for 20 min (Sigma 3–16PK centrifuge, Sigma). To precipitate alginate, an equal volume of $CaCl_2$ (1% (*w/v*)) was added to the soluble fraction, and the mixture was incubated at 4 °C overnight. Subsequently, alginate was separated by centrifugation at $5000\times g$ (Sigma 3–16PK centrifuge, Sigma) for 20 min. The laminarin extract was then separated on a 10 kDa filter using a tangential flow filtration (Millipore, Burlington, MA, USA) to remove high-molecular-weight carbohydrates and then dialyzed through a 1 kDa filter. Extracted laminarin was lyophilized and stored at 4 °C before further use.

### 2.4.2. Expression and Purification of Laminarinase *Ml*Lam17B

The recombinant *Ml*Lam17B [18] was produced as a fusion protein in a modified L-rhamnose inducible dual-tag pJOE vector [20] in *Escherichia coli* NiCo21 (DE3) fused with an N-terminal maltose binding (MBP) domain separated by a *Saccharomyces cerevisiae* ubiquitin-like protein motif (Smt3) and C-terminal hexahistidine affinity tag. The recombinant protein was purified from clarified cell lysate by maltose affinity chromatography and subsequently by nickel affinity chromatography, after proteolytic cleavage of the MBP domain with ubiquitin-like-specific protease 1 (Ulp1). The integrity and purity of the protein were analyzed by 4–15% glycine-SDS-PAGE. The protein concentration was estimated by measuring $A_{280}$ using a NanoDrop 1000 spectrophotometer (Thermo Fisher Scientific).

### 2.4.3. Enzymatic Hydrolysis of Laminarin

A solution of 20 g/L laminarin was prepared in 50 mM acetate buffer pH 4.5. The mixture was heated for 30 min while mixing with a magnetic stirrer to enhance the solubilization of laminarin. The mixture was cooled down and subsequently centrifuged. The supernatant was subjected to enzymatic hydrolysis with 10 mg of *Ml*Lam17B enzyme, performing the reaction overnight at 40 °C and 160 rpm using a PCMT thermoshaker (Biosan). The reaction was terminated by autoclavation at 1 atm for 20 min. The obtained hydrolysate was used for further analysis and fermentation experiments.

### 2.5. Carbohydrate Quantification

The seaweed biomass was lyophilized by freeze drying (Labconco, Kansas City, MO, USA) and ground to a particle size below 1 mm prior to carbohydrate analysis. The total carbohydrate content of dried *A. esculenta* was measured using a two-step sulfuric acid hydrolysis method described by van Wychen and Laurens [21]. Initially, 2.50 mL of 72% (*w/w*) sulfuric acid was added to 0.25 g of seaweed powder and incubated at 30 °C while shaking at 300 rpm (ISF1-X (Climo-Shaker), Adolf Kühner, Birsfelden, Switzerland) for 1 h. Subsequently, the acid was diluted with distilled water to 4% (*w/w*), and the samples were autoclaved at 121 °C for 1 h. The resulting hydrolysate was then centrifuged at $4000\times g$ (Sigma 3–16PK centrifuge, Sigma) for 10 min after being cooled to room temperature. The supernatant was then neutralized with 0.1 M $Ba(OH)_2.H_2O$ (0.1 M), and the neutral supernatant was collected after centrifugation at $4000\times g$ (Sigma 3–16PK centrifuge, Sigma) for 10 min. The hydrolysate was further diluted and filtered through a PTFE membrane 0.2 μm pore size filters (Pall) before analysis of monosaccharides.

Mono- and oligosaccharides were quantified using a high-performance anion exchange chromatography Dionex ICS-5000 (Thermo Fisher Scientific) system equipped with pulsed amperometric detection (HPAEC-PAD). The system was equipped with a Dionex CarboPac PA-20 column coupled with a guard column (Thermo Fisher Scientific) for monosugar analysis and a Dionex CarboPac PA200 column coupled with a guard column (Thermo Fisher Scientific) for oligosaccharide analysis. Eluents were prepared as (A) ultrapure water (Milli-Q grade), (B) 2 mM NaOH for the PA-20 column, 1 M Na-acetate in 200 mM NaOH for the PA-20 column for uronic acid analysis, or PA-200 column for oligosaccharides analysis, and (C) 200 mM NaOH. Monosaccharides were separated by isocratic elution with an eluent mixture of (A) 62.5% and (B) 37.5% for 25 min. Afterward, the column was

regenerated using an eluent mixture of (A) 50% and (C) 50% for 5 min. Uronic acids were eluted by an eluent mixture of (A) 55%, (B) 15%, and (C) 30% for 15 min. Oligosaccharides were eluted by a gradient mixture of (B) from 0 to 50% in eluent (C), while eluent (A) was set to 50% for 20 min. The column was then regenerated with an eluent mixture of (A) 50% and (B) 50% for 10 min. All analyses were performed by applying the flow rate of 0.5 mL/min and the temperature of the column compartment of 30 °C.

### 2.6. Metabolite Quantification

A high-performance liquid chromatography (HPLC) Dionex UltiMate 3000 RSLC (Thermo Fisher Scientific) system, equipped with an Aminex HPX-87H column (Bio-Rad), was used to quantify short-chain fatty acids (SCFAs), lactic acid, and ethanol. The cultivation samples were centrifuged at $3000 \times g$ (Centrifuge 5425, Eppendorf) for 10 min at RT. One milliliter of the supernatant was mixed with 20 µL of 20% (*v/v*) sulfuric acid and incubated for 15 min at 4 °C. The samples were filtered through PTFE membrane 0.2 µm pore size filters (Pall) and stored at 4 °C. Lactic acid and SCFAs were separated using 5 mM sulfuric acid at a flow rate of 0.5 mL/min, while the column compartment was kept at 40 °C. To detect and quantify ethanol, 5 mM sulfuric acid was used as the eluent, and the temperature of the column compartment decreased to 30 °C.

### 2.7. Statistical Analysis

Statistical analysis was conducted using both Excel and GraphPad Prism software version 9.4.0 (San Diego, CA USA). All measurements were taken in duplicate, and the results were presented as mean ± standard deviation. To explore significant differences in the growth behavior of the LAB consortium in the presence of various substrates, a one-way analysis of variance (ANOVA) and *T*-test were conducted in Jamovi [22], with a significance level of 0.05. Tukey's post hoc test was employed to identify any differing sample groups.

## 3. Results

### 3.1. Selection and Identification of the Bacterial Species in the LAB Consortium

The LAB consortium used was chosen based on a prescreening of isolates from food sources for the ability to utilize mannitol, as this, if consumed in excess, can cause diarrhea, abdominal pain, and flatulence [5]. Hence, mannitol was in the first screen used as the sole carbohydrate source in the MRS medium. The selected mannitol utilizing isolate was subjected to 16S rRNA gene sequencing as well as metagenome sequencing. The 16S rRNA sequencing was performed after PCR amplification and resulted in a single sequence with 99.93% sequence identity at 100% sequence coverage with a 16S rRNA gene (RefSeq NR_115605.1) from *Lactiplantibacillus plantarum* subsp. *plantarum* (JCM 1149) ATCC 14917, isolated from pickled cabbage [23,24].

Metagenome analysis, implementing de novo genome assembly, revealed the presence of a microbial consortium composed of three related *L. plantarum* strains and one *L. brevis* strain. The *L. plantarum* strains dominated the consortium (93.65% abundance) with a relative abundance distribution of 45.41% (closest related deposited strain *L. plantarum* subsp. *plantarum* [25]), 41.69% (closest related candidate *Lactobacillus* sp. LSI2-1 [26]), and 6.55% (closest related deposited candidate *L. japonicus* [27]). Moreover, *Lactobacillus* sp. LSI2-1 is previously reported to be taxonomically close to *L. plantarum* subsp. *plantarum* ATCC 14917 with 100% 16S rRNA sequence similarity at 100% sequence coverage [25], and *L. japonicus* is reclassified as *L. plantarum* subsp. *plantarum* [27], explaining the uniform data from 16S rRNA-gene sequencing. The relative abundance of the strain related to *L. brevis* [25] was minor, estimated at 6.35% (Figure 2).

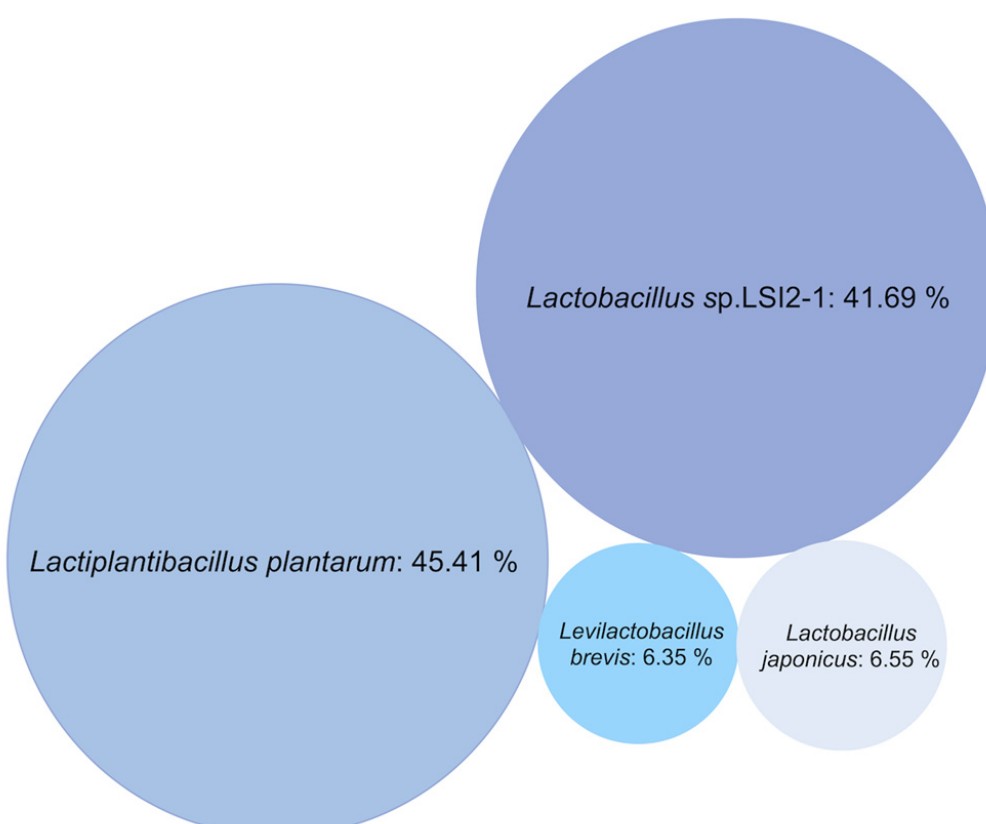

**Figure 2.** Bubble chart representing the relative abundance of bacterial strains in the LAB consortium.

*3.2. Growth on Monosaccharides as Sole Carbohydrate Source in MRS, and the Relation to Brown Seaweed Polysaccharides*

The selected consortium was subjected to growth screening in liquid MRS broth to verify its ability to grow on monosaccharides, which can be found as building blocks in brown seaweed polysaccharides. The growth was estimated by measuring $OD_{600}$ and carbohydrate consumption. As shown in Figure 3, the highest cell density was observed when the LAB consortium was grown in MRS supplemented with 20 g/L glucose (the monosugar in the polymer laminarin), reaching a final $OD_{600}$ of $15.55 \pm 0.35$ after 24 h (Figure 3a). Somewhat lower cell densities, with a maximum $OD_{600}$ of $7.9 \pm 0.28$ ($p < 0.05$ comparing to glucose), were observed in the presence of mannitol (Figure 3b), which is a free sugar alcohol in the brown seaweeds, as well as the capping sugar alcohol in a fraction of the laminarin polymers. Moreover, galactose (a monomer component in fucoidan) resulted in an $OD_{600}$ of $6.92 \pm 0.42$, in the same range (Figure 3c). The use of other monosaccharides as the carbohydrate source resulted in lower $OD_{600}$ at the corresponding concentration of the carbohydrate source. For example, mannose resulted in an $OD_{600}$ of $3.2 \pm 0.53$ (Figure 3d) ($p < 0.05$ comparing to glucose), while MRS supplemented with xylose (a minor component in fucoidan) displayed a long lag phase prior to growth, reaching an $OD_{600}$ of $2.54 \pm 0.03$ ($p < 0.05$ comparing to glucose) (Figure 3e) after 72 h cultivation. The use of xylose as a substrate may also be a consequence of the growth of the *L. brevis* strain, present in minor relative abundance in the consortium, as this species has previously been shown to utilize xylose efficiently [28]. The LAB consortium was not able to utilize fucose (from fucoidan), mannuronic acid, or guluronic acid (the building blocks of alginate). In conclusion, glucose and mannitol were almost completely depleted in the cultivations, while incomplete utilization was observed for the other carbohydrates.

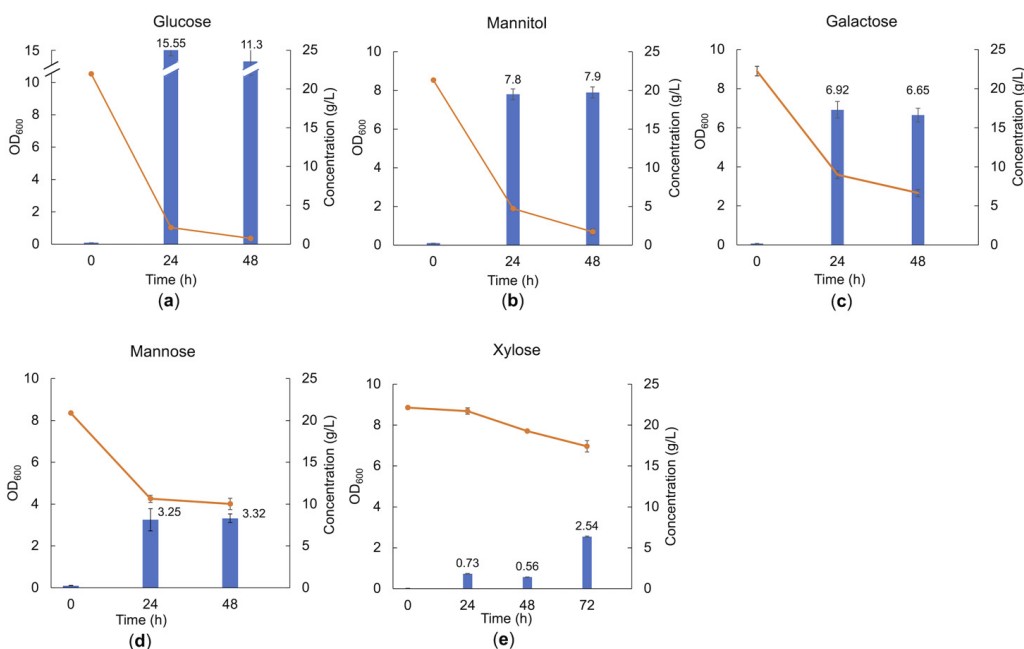

**Figure 3.** Cell density and carbohydrate consumption of the LAB consortium in the presence of glucose (**a**), mannitol (**b**), galactose (**c**), mannose (**d**), and xylose (**e**) after 48 h or 72 h batch cultivation in MRS medium supplemented with 20 g/L of the respective substrates. Blue bars represent the optical density at 600 nm ($OD_{600}$); orange line represents sugar concentration in the medium; and error bars show the standard deviation between duplicate measurements.

In co-cultivations, with both glucose and mannitol in the medium, the consortium preferably utilized glucose, showing glucose catabolite repression, reaching an $OD_{600}$ of $8.4 \pm 0.03$ with an initial glucose concentration of 12 g/L (Figure 4). Only a small amount of mannitol (0.9 g/L) was used before reaching the stationary phase, and in total, 2.7 g/L of mannitol was consumed over 72 h of cultivation. The specific growth rates in the presence of individual substrates (glucose and mannitol) were calculated as 0.66 $h^{-1}$ in the presence of glucose and 0.56 $h^{-1}$ in the presence of mannitol. However, the specific growth rate throughout the co-fermentation of glucose and mannitol was obtained at 0.65 $h^{-1}$, similar to the specific growth rate in the presence of only glucose.

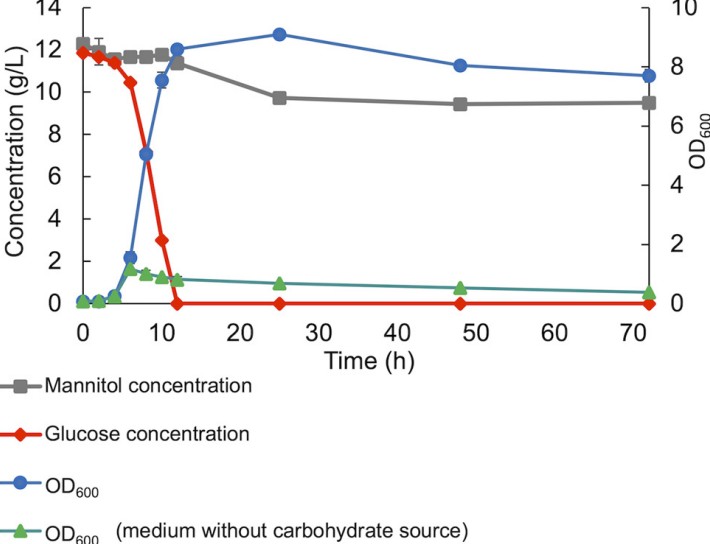

**Figure 4.** Co-fermentation of glucose and mannitol by the LAB consortium throughout anaerobic batch cultivation. Standard deviation of duplicate samples is shown.

### 3.3. Growth of the LAB Consortium on Polysaccharides or Oligosaccharides

Utilization of the brown seaweed polysaccharides, alginate, laminarin, and fucoidan was tested separately as carbohydrate sources in MRS, but the bacteria did not show any growth on any of the polymeric substrates.

Hence, the ability of the LAB consortium to grow on a selection of oligosaccharides was investigated with a focus on glucose-containing oligosaccharides connected by different linkages and of different degrees of polymerization (DP).

The growth of the LAB consortium on β-glucan oligosaccharides was tested in an MRS broth base individually supplemented with the laminari-oligosaccharides of DP2-7 (O-LAM2-7). The cell density was monitored after 24 h anaerobic cultivation in the presence of 5 g/L substrate at 37 °C. As illustrated in Figure 5, in the presence of glucose, the $OD_{600}$ increased to $3.7 \pm 0.1$, while a slightly higher OD was reached in the presence of laminaribiose (O-LAM2) ($OD_{600}$ of $4.3 \pm 0.18$) and laminaritriose (O-LAM3) ($OD_{600}$ of $4.64 \pm 0.13$). The pH dropped to almost 4.4 during the cultivations using these substrates. A slight increase in $OD_{600}$ was also observed in the presence of laminaritetraose (O-LAM4) ($OD_{600}$ of 1.1) with a pH drop to 5.9. Negligible growth was observed in the presence of laminaripentaose (O-LAM5), with no pH drop (pH remaining at 6.5, which was the initial pH of the medium), and no growth was observed in the presence of laminarihexaose (O-LAM6) and laminariheptaose (O-LAM7), indicating a DP4 polymerization limit for laminari-oligosaccharide utilization of the consortium.

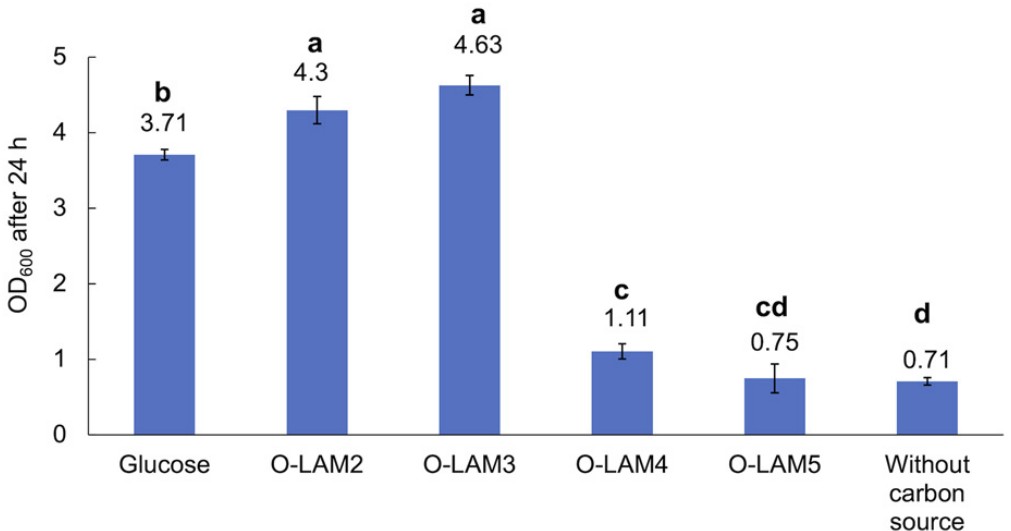

**Figure 5.** Growth of the LAB consortium in the presence of glucose and laminari-oligosaccharides O-LAM2-5. O-LAM2, O-LAM3, O-LAM4, and O-LAM5 represent laminaribiose, laminaritriose, laminaritetraose, and laminaripentaose, respectively. Standard deviation between duplicates is shown as error bars. Significant ($p < 0.05$) differences are indicated by letters (a–d).

The laminari-oligosaccharides are connected by β-1,3-linkages. To evaluate if the linkage type is of importance, two other glucose-containing disaccharides were also tried as carbohydrate sources, in particular maltose (a disaccharide with glucose building blocks connected by an α-1,4 linkage) and sucrose (a disaccharide composed of α-D-glucopyranosyl-(1,2)-β-D-fructo-furanoside). The LAB consortium consumed both maltose and sucrose, reaching an $OD_{600}$ of $10.61 \pm 0.81$ and $8.4 \pm 0.14$, respectively, after 24 h of cultivation in MRS broth base supplemented with 20 g/L of the respective carbohydrate substrate. This showed that disaccharides connected by various linkages could be consumed (Figure 6). In addition, the growth of the LAB consortium on xylo-oligosaccharides was investigated; however, these substrates were not consumed by the consortium within the time frame of the experiments.

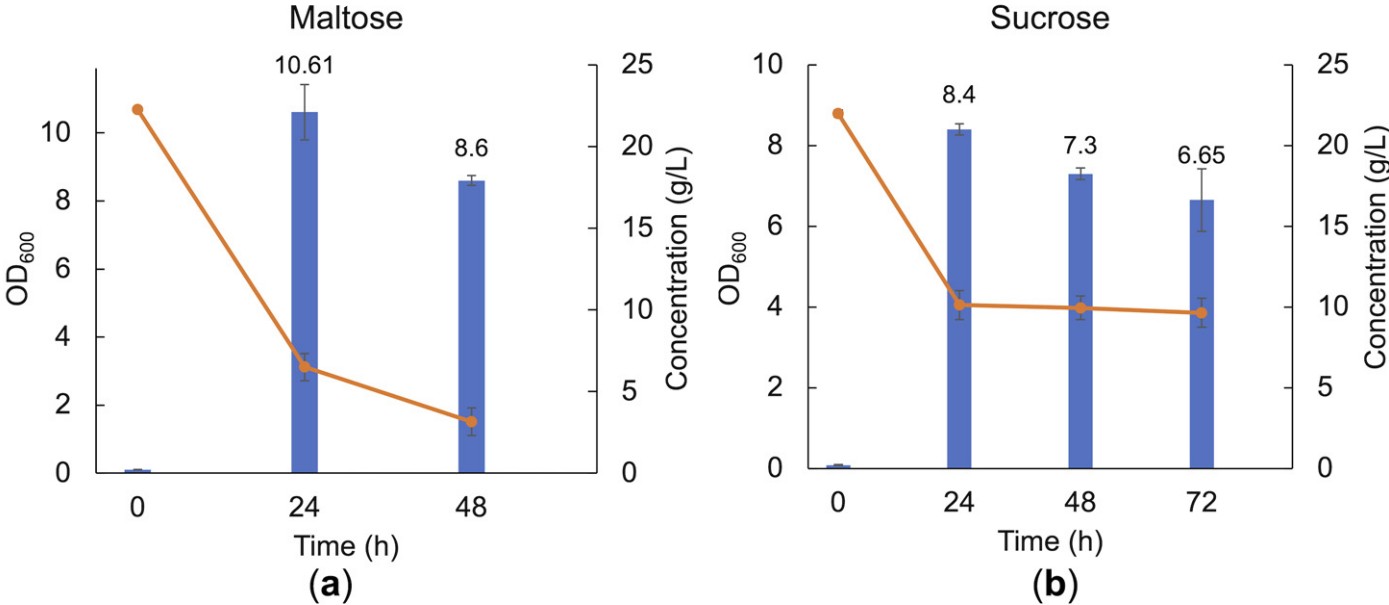

**Figure 6.** Cell density and carbohydrate consumption of the LAB consortium in the presence of maltose (**a**) and sucrose (**b**) after 24 h, 48 h, and 72 h batch cultivation in MRS broth base supplemented with 20 g/L of the respective carbohydrate substrate. Error bars represent the standard deviation of duplicates.

### 3.4. Co-Fermentation of Laminari-Oligosaccharides by the LAB Consortium

#### 3.4.1. Laminari-Oligosaccharides Production from a *L. hyperborea* Extract

A mixture of oligosaccharides was prepared by enzymatic hydrolysis of laminarin extracted from *L. hyperborea* using the in-house-produced laminarinase *Ml*Lam17B (Figure S2). HPAEC-PAD analysis of the hydrolysate demonstrated that it contained a mixture of laminari-oligosaccharides, glucose, and mannitol (Table 1). The mannitol in the hydrolysate likely originates from laminarin chains that contain a terminal mannitol moiety at the reducing end [1], called M-chain laminarins.

**Table 1.** Carbohydrate composition of the hydrolysate obtained after enzymatic hydrolysis of laminarin, extracted from *L. hyperborea*, and hydrolyzed with *Ml*Lam17B.

| Carbohydrate Composition (% DW) | | | | |
|---|---|---|---|---|
| Glucose | Mannitol | Laminaribiose (O-LAM2) | Laminaritriose (O-LAM3) | Laminaritetraose (O-LAM4) |
| 0.38 | 2.17 | 16.21 | 29.77 | 11.52 |

#### 3.4.2. Co-Fermentation of Laminari-Oligosaccharides

Co-fermentation of the laminari-oligosaccharides present in a hydrolysate by the LAB consortium in the MRS broth medium is shown in Figure 7a. The low amount of glucose present in the hydrolysate was consumed before any decrease in the concentration of the oligosaccharides could be observed. Thus, to evaluate if glucose repression was occurring, the hydrolysate was supplemented with additional glucose to a final concentration of 2.5 g/L (Figure 7b). The supplementation of glucose clearly showed that glucose and the laminari-oligosaccharides were simultaneously utilized during the exponential growth phase, indicative of separate transporter systems. Hence, it can be concluded that the presence of glucose did not repress the utilization (or uptake) of the oligosaccharides in the hydrolysate.

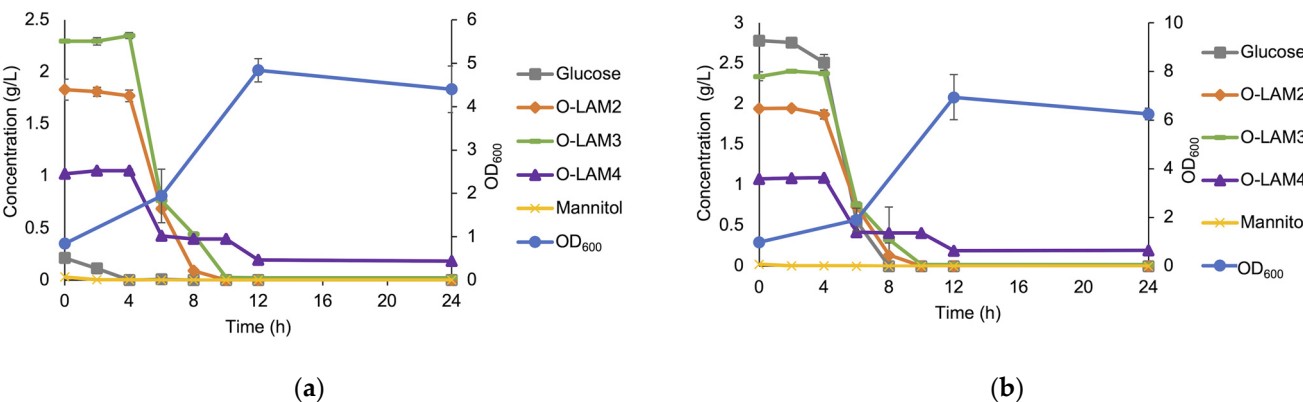

**Figure 7.** The growth of the LAB consortium in the medium supplemented with laminarin hydrolysate (5 g/L) (**a**) and hydrolysate (5 g/L) with additionally added glucose to a final concentration of 2.5 g/L (**b**). Error bars represent the standard deviation of duplicates.

Even though the hydrolysate contains less glucose than mannitol, the concentration of glucose in the medium is higher. This may be attributed to the sugar content of the components of the medium, such as yeast extract [29]. Mannitol, which was present in a negligible amount in the medium, was also consumed along with glucose before the LAB consortium utilizes oligosaccharides (Figure 7a).

### 3.5. Direct Fermentation of Fresh Frozen A. esculenta

The possibility of direct fermentation of fresh frozen, minced brown seaweed of the species *A. esculenta* was evaluated, as a potential preservation method of the seaweed.

To estimate the number of colonies in the starting culture, the relation between optical density and CFU/mL was calculated as CFU/mL= $3 \times 10^8$ OD$_{600}$ $- 1 \times 10^8$ ($R^2$ = 0.9493). An inoculum containing $3 \times 10^6$ cells/g of the seaweed slurry was added to the reactors, prepared for fermentation with the LAB consortium. Duplicate reactors were prepared for the fermentation of autoclaved and non-autoclaved seaweed biomass. The inoculum was not added to the control reactors (one with autoclaved and one with non-autoclaved seaweed slurry) prepared to evaluate any possible fermentation by endogenous microbiota.

Carbohydrate quantification of the dried *A. esculenta* seaweed, which underwent a two-step acid hydrolysis process, revealed that it comprises approximately 29% carbohydrates, consisting of around 12% neutral sugars (of which 6% was mannitol) and approximately 17% uronic acids that are the primary constituents of alginate (Table S1). The analysis of carbohydrates in the soluble fraction of the seaweed slurries, including both autoclaved and non-autoclaved reactors, indicated that only mannitol was released into the liquid suspensions. The slight increase in the concentration of mannitol at the beginning is judged to be due to the solubilization of mannitol by the heat and pressure applied [30]. In both inoculated conditions, the CFU increased up to approximately $3 \times 10^7$ CFU/mL ($p < 0.05$) after 72 h of fermentation, but the CFUs decreased significantly after 7 days, after depletion of the mannitol that served as the carbohydrate source.

The growth behavior after inoculation was very similar in the autoclaved and non-autoclaved seaweed slurry reactors (Figure 8a). The initial pH in the autoclaved reactor was approximately pH 6.2 (Figure 8b), while the corresponding initial pH in the non-autoclaved reactor was 6.5. This pH difference at the beginning of the fermentation is likely due to the release of carbohydrates and minerals into the seaweed suspension during autoclaving. In inoculated reactors, after approximately a 3 h lag phase, the pH decreased sharply (indicating acid production), reaching pH 4.3 in non-autoclaved reactors and pH 4.7 in autoclaved reactors within 18 h, followed by a gradual decrease until the end of the fermentation, reaching a final pH below pH 4 (pH 3.8 in autoclaved and pH 3.9 in non-autoclaved reactors). Gas production was not observed during the 7-day fermentation, neither in autoclaved nor in non-autoclaved seaweed slurry.

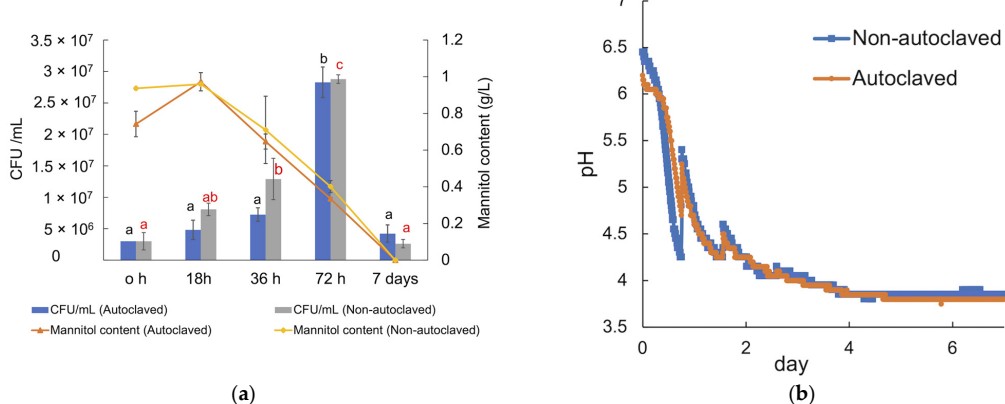

| | |
|---|---|
| (**a**) | (**b**) |

**Figure 8.** (**a**) Carbohydrate content and CFU/mL of the LAB consortium in the autoclaved and non-autoclaved reactors over seven days of fermentation at 37 °C. (**b**) pH curve from the fermentation of seaweed biomass, obtained from a gas endeavor system; orange curve represents the pH drop in autoclaved reactors, and the blue curve indicates the pH drop in non-autoclaved reactors. The experiments were performed in duplicates, and the error bars represent the standard deviation of duplicates. Significant ($p < 0.05$) differences are indicated by black letters (a–b) for autoclaved samples and by red letters (a–c) for non-autoclaved samples.

In the non-inoculated reactor with autoclaved seaweed slurry, run to monitor the growth of endogenous microbiota, only a slight drop in pH to pH 5.7 was observed while mannitol was consumed completely. In this case, the seaweed slurry became slimy and foamed, and an unpleasant smell was noticed after opening the reactor. Metagenome analysis of the corresponding seaweed sample revealed that the seaweed contained bacteria (relative abundance 96%), fungi (relative abundance 2.8%), and minor amounts of archaea, viruses, and Eukaryota (relative abundance 1.4%). A mixture of Gram-positive spore-forming bacteria strains (93.76% from the genus *Bacillus* and 6.23% from the genus *Paenibacillus*) was present in the seaweed slurry (Figure 9), which could not be eliminated by autoclaving, and incubation at 37 °C of autoclaved seaweed slurry resulted in their growth and, hence, spoilage of the seaweed. Similar spoilage was also observed for non-autoclaved seaweed slurry without LAB inoculation. However, metagenome analysis of the latter was unsuccessful due to insufficient amounts of DNA for sequencing.

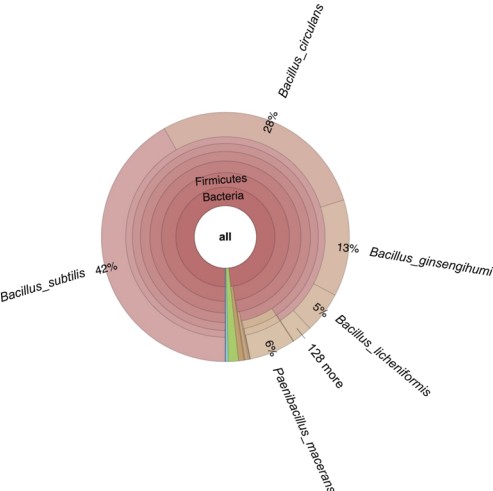

**Figure 9.** Krona-plot obtained from metagenome analysis of fermented seaweed from the autoclaved reactor without LAB consortium inoculation, performed for 7 days at 37 °C. The green area represents Eukaryota (relative abundance of 1%), and blue and purple areas represent viruses (relative abundance of 0.4%) and archaea (relative abundance of 0.01%), respectively.

### 3.6. Analysis of Metabolites Profile

Lactic acid was the main metabolite produced by the LAB consortium when cultivated in an MRS medium with all variants of the carbohydrate sources (monosaccharides and oligosaccharides), as well during seaweed fermentation (Table 2). Maximum production of lactic acid (21.91 ± 0.32 g/L) was observed in cultivations run in the presence of approximately 22 g/L glucose, after 48 h anaerobic cultivation in MRS (Figure 10a and Table 2).

**Table 2.** Lactic acid production of the LAB consortium grown in the presence of monosaccharides, a mixture of glucose and mannitol, crude laminari-oligosaccharides with /without added glucose, and during fermentation of raw *A. esculenta*.

| Substrate | Lactic Acid (g/L) |
| --- | --- |
| Glucose (21.93 ± 0.14 g/L) | 21.91 ± 0.32 |
| Mannitol (21.32 ± 0.24 g/L) | 21.38 ± 0.12 |
| Maltose (22.26 ± 0.05 g/L) | 22.21 ± 0.64 |
| Galactose (22.25 ± 0.62 g/L) | 19.22 ± 0.31 |
| Sucrose (21.98 ± 0.23 g/ L) | 16.1 ± 0.24 |
| Glucose + mannitol (12.12 ± 0.49 g/L glucose + 11.87 ± 0.14 g/L mannitol) | 14.21 ± 0.68 |
| Mannose (20.86 ± 0.14 g/L) | 11.13 ± 0.51 |
| Crude laminari-oligosaccharides (5.36 ± 0.1 g/L) + glucose (2.78 ± 0.02 g/L) | 7.65 ± 0.02 |
| Crude laminari-oligosaccharides (5.15 ± 0.1 g/L) | 4.98 ± 0.07 |
| Xylose (21.13 ± 0.1 g/L) | 3.29 ± 0.08 |
| Fermentation, raw *A. esculenta*, autoclaved (1 g/L mannitol) | 1.03 ± 0.03 |
| Fermentation, raw *A. esculenta*, non-autoclaved (1 g/L mannitol) | 1.03 ± 0.01 |

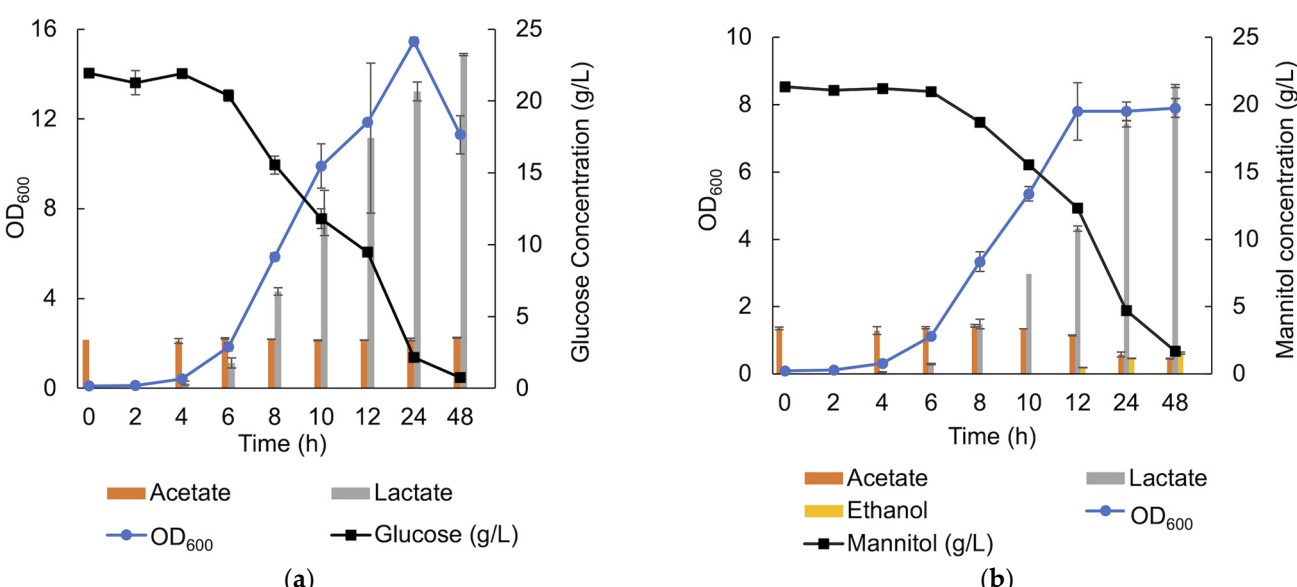

**Figure 10.** SCFA, lactic acid and ethanol production in the presence of 20 g/L of glucose (**a**) and 20 g/L of mannitol (**b**) in MRS under anaerobic condition.

In MRS containing approximately 21.3 g/L mannitol, lactic acid production was observed from an early stage of the growth and continued throughout the cultivation, reaching a maximum of 21.38 ± 0.12 g/L. However, when the stationary phase was entered, acetate (From MRS) was consumed (2.25 ± 0.07 g/L acetate consumption), and ethanol production (1.6 ± 0.1 g/L) was observed (Figure 10b), which was not the case using glucose as the sole carbon source. A small peak corresponding to succinate was also observed after

8 h cultivation but was not quantifiable. In the presence of xylose, acetate was produced with a final concentration of $1.23 \pm 0.04$ g/L over 72 h of cultivation.

During the co-fermentation of glucose and mannitol in MRS, lactic acid was the only metabolite detected. In this case, due to glucose catabolite repression, mannitol consumption may have been too low to result in the detectable production of the previously identified metabolites (succinate and ethanol). The same could be generalized for all substrates that contained mannitol because in the cultivation on laminari-oligosaccharides hydrolysate and in the cultivation on seaweed slurry, lactic acid was the only detectable metabolite.

The data presented in Table 2 show that the production yield of lactic acid (Lactic acid/substrate (*w/w*)) is nearly 100% for glucose, mannitol, maltose, and laminari-oligosaccharides. A lower yield was observed when galactose, sucrose, mannose, and a mixture of glucose and mannitol were used as the substrate. The lactic acid production yield was approximately 86% in the presence of galactose, nearly 53 % in the presence of mannose, about 73% in the presence of sucrose, and almost 60% when a mixture of glucose and mannitol was used as the substrate. The lowest yield of approximately 16% was observed in the presence of xylose substrate.

## 4. Discussion

This study proved the potential of a LAB consortium in fermenting the brown seaweed *A. esculenta*. LAB fermentation is a promising preservation technique for seaweed in new food markets. Mannitol and laminari-oligosaccharides from the brown algae polysaccharide laminarin were verified as substrates for the consortium, with some potential prebiotic properties, indicated by the growth of the LAB and metabolite profile during mannitol utilization.

The genomic analysis of the consortium revealed that it consisted of four strains, *L. plantarum* (closely related to *L. plantarum* strain (JCM 1149) ATCC 14917 *L.* sp. LSI2-1, *L. japonicus*, and a relatively minor amount of *L. brevis*. The strain *Lactobacillus* sp. LSI2-1 is closely related to *L. plantarum* subsp. *plantarum* ATCC 14917, which, according to previous characterization, is able to utilize glucose, mannitol, and xylose. Moreover, the strain did not show any gas production in the presence of glucose [22], in accordance with our data. *L. japonicus* is closely related to *L. plantarum* subsp. *plantarum*, and this bacterium is a DL-forming LAB, isolated from plants with the ability to ferment pentoses [27]. Xylose fermentation is also a feature of *L. brevis* strains [28], and xylose was observed to be fermented slowly in this study.

This study demonstrates that the investigated LAB consortium can consume most monosugars, which are the building blocks of the polysaccharides in brown seaweed, including glucose, mannitol, galactose, xylose, and mannose but except for fucose, which is the building blocks of fucoidan.

Glucose and mannitol were the most favorable substrates for the consortium and were completely consumed over the 48 h anaerobic cultivation period, being the sole carbohydrate source in the medium. However, in co-fermentation, glucose was the preferred substrate, and only after the depletion of glucose, mannitol was consumed by the bacteria. In the co-fermentation of glucose and mannitol at high concentrations of the carbohydrate sources, the growth reached the stationary phase by mainly consuming glucose, and mannitol was only used by the bacteria for maintenance. The entry of the stationary phase could be due to some limitation in the medium or by the low pH of the medium reached at this stage due to the lactic acid production. However, at a lower initial concentration of glucose, the bacteria consumed mannitol for growth after glucose depletion. The metabolite profile of the culture supernatants indicated that the consortium employed two different pathways in the utilization of glucose and mannitol since in the presence of glucose, lactic acid is the only metabolite, while in the presence of mannitol, in addition to lactic acid, ethanol and succinate are produced while acetate is consumed. It appears that in the presence of glucose, the consortium uses the Embden–Meyerhof–Parnas pathway (EMP pathway). In this pathway, pyruvate, the final product of the EMP pathway, is directly reduced to lactate by

NAD$^+$-dependent lactate dehydrogenase, thereby reoxidizing NADH, which was produced in the early glycolytic steps [31]. This pathway was used by *L. plantarum* WCFS1 [32]. For mannitol utilization, McFeeters and Chen [33] proposed a pathway for *L. plantarum* ATCC 14917 and *L. plantarum* C11 in which acetate was converted to ethanol when mannitol was used as the substrate in the fermentation. They also showed that in mannitol fermentation, citrate can be converted to succinate and ethanol. Since the MRS medium used in this study contained ammonium citrate, the small amount of succinate in the cultivation broth is likely due to the conversion of citrate to succinate. Yang and co-authors [34] suggested a pathway for *L. plantarum* NF92, for mannitol utilization, showing that lactate is the major metabolite in early growth with the production of negligible amount of ethanol, while in the late growth phase, ethanol is produced. Therefore, it can be concluded that in the presence of glucose and mannitol, mainly *L. plantarum* strains proliferate.

In addition to the ability to consume monosaccharides, the LAB consortium proved an ability to utilize short-chain glucan oligosaccharides with various linkage types. As was demonstrated, the LAB consortium could utilize β-laminari-oligosaccharides derived from laminarin and other oligosaccharides that do not originate from seaweed biomass, such as maltose (a disaccharide made of α-1,4 linked glucose moieties) and sucrose (a disaccharide composed of α-D-glucopyranosyl-(1,2)-β-D-fructo-furanoside). There are many studies showing that short oligosaccharides of α- and β-glucans derived from terrestrial biomass can be catabolized by lactic acid bacteria [35]. However, only a few studies show the utilization of oligosaccharides derived from marine biomass by LAB. In this area, Jin et al. [36] showed that various lactic acid bacteria species, including *L. plantarum*, encode specific transporter systems for the utilization of α-gluco-oligosaccharides derived from the brown seaweed *Laminaria japonicus*. However, there is a lack of evidence regarding the growth of LAB on β-gluco-oligosaccharides derived from seaweed. In this study, it was shown that the LAB consortium could utilize β-1,3 linked laminari-oligosaccharides (DP2-4) derived from the brown seaweed *L. hyperborea*. This indicates that short laminari-oligosaccharides have the potential to be utilized. No growth was observed in the presence of polysaccharides available in the *A. esculenta* brown seaweed.

Fermentation with LAB is a well-established preservation method and is proven to enhance the taste, texture, and shelf life of food products by releasing metabolites that lower the pH, hence preventing pathogenic and spoilage microorganisms from growing. In this study, the LAB consortium proved to be able to ferment the brown seaweed *A. esculenta*, which is the second most cultivated seaweed species in Europe. The pH decreased to approximately pH 3.8 (from pH 6.2 in autoclaved reactors and pH 6.5 in non-autoclaved reactors) during the fermentation, while the consortium catabolized mannitol (the only carbohydrate available in the fermentation broth), which, if consumed in excess, may have a laxative effect and may cause diarrhea [37]. The pH drop via fermentation assures that the endogenous spoiling bacteria and fungi are prevented from growing since they only grow in an environment with a pH above pH 4.6 [38–40]. This shows the importance of seaweed fermentation since the microbiota of *A. esculenta* is still unknown and could contain potential pathogenic microorganisms [41]. As was demonstrated in this study, fermentation with the endogenous microbiota (seaweed samples without any added LAB inoculum) for seven days resulted in spoilage, where the seaweed became slimy with an unpleasant smell while the pH dropped to pH 5.7. The metagenome analysis of the seaweed fermented with endogenous bacteria showed the presence of spore-forming bacteria [42] that cannot be removed by autoclaving, and fermentation of the seaweed without adding a LAB inoculum resulted in spoilage of the seaweed. However, by adding a LAB inoculum to the samples, the spoilage of the seaweed was prevented.

## 5. Conclusions

The LAB consortium investigated in this study consists of three *Lactiplantibacillus plantarum* strains and a minor relative abundance of an *Levilactobacillus brevis* strain. This consortium showed substantial ability in fermenting monosaccharides, which are the

building blocks of the carbohydrate polymers, excluding fucose, mannuronic acid, and guluronic acid. Additionally, the consortium was also proven to utilize maltose, sucrose, and laminari-oligosaccharides (DP2-4) derived from laminarin. Lactic acid was the major metabolite produced; however, when mannitol was consumed, succinate and ethanol were produced while acetate (from the medium) was consumed. In xylose catabolism, in addition to lactic acid, acetate was also produced. The consortium could successfully directly ferment the brown seaweed *Alaria esculenta*, lowering the pH to pH 3.8 (by producing lactic acid), while utilizing mannitol, the only carbohydrate released into fermentation broth. Fermentation with seaweed's endogenous bacteria resulted in seaweed spoilage. This suggests the need for the preservation of the seaweed *A. esculenta*, and fermentation proved to be a promising method to prolong its shelf life.

**Supplementary Materials:** The following supporting information can be downloaded at: https://www.mdpi.com/article/10.3390/fermentation9060499/s1, Figure S1: Agarose gel electrophoresis of 16S rRNA gene, amplified with universal fD1 and rP2 primers. GeneRuler 1 kb DNA Ladder (Thermo Fisher Scientific) was used as the ladder; Figure S2: *Ml*Lam17B purity and integrity determination with SDS-PAGE. Precision Plus Protein Dual Color Standards (Bio-Rad) molecular-mass marker was used to estimate the molecular weight of protein; Table S1: Composition analysis of *Alaria esculenta* after lyophilization and acid hydrolysis.

**Author Contributions:** Conceptualization, G.Ó.H. and E.N.K.; methodology, L.A., M.J., A.J., M.D.C., J.A.L.-P., S.L. and M.N.; data curation, validation, and visualization, L.A.; formal analysis and investigation, L.A., M.J., A.J., M.D.C., J.A.L.-P., S.L., M.N. and E.N.K.; resources, E.N.K.; writing—original draft preparation, L.A.; writing—review and editing, L.A., M.J., M.D.C., A.J., J.A.L.-P., S.L., G.Ó.H. and E.N.K.; supervision, E.N.K.; project administration, E.N.K.; funding acquisition, E.N.K. All authors have read and agreed to the published version of the manuscript.

**Funding:** This research was funded by the Era-net ProSeaFood (Era-net SusFood2, grant agreement no. 727473) supported by Formas, the Eurostars project SeaPro, supported by Vinnova, grant no. 201-03368, and the Era-net Cofund BlueBio MariKat (New Catalytic Enzymes and Enzymatic Processes from the Marine Microbiome for Refining Marine Seaweed Biomass) project (grant no. 2019-02359), from the Swedish Research Council Formas.

**Institutional Review Board Statement:** Not applicable.

**Informed Consent Statement:** Not applicable.

**Data Availability Statement:** The data are available on request from the corresponding author.

**Conflicts of Interest:** The authors declare no conflict of interest.

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
