# Peer review of "Fermentation of the Brown Seaweed Alaria esculenta by a Lactic Acid Bacteria Consortium Able to Utilize Mannitol and Laminari-Oligosaccharides"

_fermentation, doi:10.3390/fermentation9060499_

Round 1

Reviewer 1 Report

The paper entitled ‘’Fermentation of the Brown Seaweed Alaria esculenta by a Lactic Acid Bacteria Consortium able to Utilize Mannitol and Laminari-Oligosaccharides’’ treats a relative uninvestigated and interesting topic. The methods used in this study are provided in sufficient detail. However, there are some adjustments that need to be made:

Point 1 (Material and Methods):

1.1. The flowchart representing the fermentation experiment (Figure 1) is relative unclear and may give rise to erroneous interpretations. Moreover, the information provided in text (lines 141 - 142 and 149 - 151 respectively) does not seem to be overlapped on the information from Figure 1.

1.2. Please explain how ‘’the fermentation with seaweed’s endogenous microbiota was performed by fermenting seaweed slurry with pre-treatment (with autoclaving) without added inoculums’’ (lines 149 – 151)? The endogenous microbiota was not killed by autoclaving? What temperature was reached by autoclaving?

1.3. Please explain why the statement from lines 96-97 (‘’laminari-oligosaccharides from L. digitata were purchased from Megazyme‘’) and 164-165 (‘’laminari-oligosaccharides were produced by enzymatic hydrolysis‘’) are not convergent.

1.4. What is the origin of Laminaria hyperborea from which laminarin was extracted (line 169)? Please explain why laminarin was extracted from L. hyperborea and not from the studied seaweed Alaria esculenta. This could be a critical point of the research.

1.5. Although ‘’the seaweed species Alaria esculenta (L.) Grev. biomass was obtained fresh frozen’’ (line 93), to line 197 is discussed about ‘’dried Alaria esculenta’’. Nothing it was previously mentioned about drying of this seaweed.

1.6. The information referring to the equipments used (autoclave, reactor, incubator, centrifuge etc.), such as type and producer are missing.

Point 2 (Results):

2.1. Figures 3 (a-e) could be aggregated in a single figure, a general overview on the topic and a comprehensive understanding of the carbohydrate consumption by the LAB consortium being this way provided.

2.2. The specific growth rates (line 292) are discussed but nothing was mentioned previously about their calculation.

2.3. Referring to the fermentation on the carbohydrate available in brown seaweed (point 2.3.1) it was mentioned that the bottles were incubated at 37°C (line 135). Co-fermentation of glucose and mannitol by the LAB consortium took place throughout anaerobic batch cultivation (line 298). The aerobiosis / anaerobiosis conditions were mentioned neither to Materials and methods nor with reference to growth of LAB on monosaccharides as sole carbohydrate source in MRS. Please explain unitary and clearly the condition of fermentation.

2.4. Please explain how was chosen the glucose concentration when it was used as sole carbohydrate source in MRS respectively in co-cultivations, with both glucose and mannitol in the medium.

2.5. Please explain how was chosen the ratio glucose / mannitol in co-cultivation experimental. According to data from Table S1, the amount of mannitol in composition of Alaria esculenta is almost double than the concentration of glucose.

2.6. Laminari-oligosaccharides produced by enzymatic hydrolysis were written in small letters, i.e. O-Lam (Table 1). In Figure 7, their presence in hydrolysate was written in capital letters. In this last case I suppose that these laminari-oligosaccharides could be considered as those purchased from Megazyme. Please explain / correct, so that to avoid any confusion.

2.7. Please explain why were studied both the growth of the LAB consortium in MRS broth base individually supplemented with the laminari-oligosaccharides (point 3.3) and co-fermentation of the laminari-oligosaccharides present in hydrolysate by the LAB consortium, also in the MRS broth medium (point 3.4.2), the more so as these oligosaccharides were not obtained from A. esculenta.

2.8. Please explain why in Table 1 is mentioned mannitol as constituent of the hydrolysate but this one is missing in Figure 7.

2.9. The experimental design of the direct fermentation of fresh frozen A. esculenta (point 3.5, rows 362 - 367) should be clearly presented and in line with information provided to Materials and methods. This information should not be redundant.

2.10. Figure S1 is not mentioned in manuscript. This Figure is missing in Supplementary material.

2.11. Table S1 is not mentioned in manuscript. Nothing is mentioned about Alaria esculenta lyophilization, those composition is detailed in Table S1.

Point 3 (Discussion):

3.1. Line 521: The statement ‘’laminari-oligosaccharides (DP2-4), derived from the brown seaweed Alaria esculenta’’ is in contradiction with the previously mentioned origin of the laminari-oligosaccharides.

3.2. Line 522: ‘’short laminari-oligosaccharides has the potential to be utilized as prebiotics’’. In my opinion, in order to sustain this statement is needed to correlate the presence / concentration of the laminari-oligosaccharides with the LAB viability and growth.

3.3. Although determination of short chain fatty acids (SCFA) as metabolites of seaweed fermentation was described (point 2.6), nothing about their concentration in different experimental variants was mentioned.

3.4. A stronger connection between the growth of the LAB consortium in the presence of different monosaccharides and oligosaccharides and fermentation of Alaria esculenta respectively should be underlined. Thus, the relevance of the stages of experimental until Alaria esculenta fermentation should be more deeply explained (i.e. what is the relationship between fermentation of maltose and sucrose respectively by LAB consortium and the aim of the study, taking into account the chemical composition of the seaweed, as it was presented by authors?).

3.5. Additional references related to Alaria esculenta could be useful in interpreting the experimental data.

Point 4 (Conclusion):

Please explain what results sustain the statement referring to improvement of the nutritional content of A. esculenta by fermentation.

The spelling should carefully checked throughout the manuscript.

Reviewer 2 Report

 This work is very interesting. Well designed and presented. The authors had evaluated the potential of a LAB consortium consisting of  three strains of Lactiplantibacillus plantarum and one Levilactobacillus brevis strain in fermenting the brown seaweed Alaria esculenta by growing the consortium on fresh frozen seaweed biomass which was either or not pretreated. Besides that, laminari-oligosaccharides were produced from laminarin as value-added products, and their potential prebiotic properties was evaluated by fermentation implementing the LAB consortium. I just have some minor comments

1.       Fig 1is misleading regarding the fermentation with seaweeds endogenous micro biota after autoclaving ! please clarify how this could be achieved as indicated from the figure or correct

2.       Does the authors performed fermentation of raw pretreated algal biomass (any pretreatment) with the bacterial LAB consortium???please show the results if any

3.       Fig 5 clarify the OLAM 2-7 for the respective reader

4.       Correct references at Line 241 and 381-382 as indicated Error! Reference source not found

5.       Correct formatting of Lactobacillus and families names at reference 24

6.       Try to enlarge all figures presented in this MS as much as possible for more clarification and remove the outer lines

AAll scientific names at Fig 9 should be corrected to italic font

Round 2

Reviewer 1 Report

The results as are presented in the revised version of the manuscript are thoroughly interpreted and substantial too.

Author Response

Dear reviewer;

Thank you so much for your nice feedback on the revised version of the manuscript.

The updated version of the manuscript is attached.

Best wishes

Leila
